# The interplay of economic shocks and cultural practices in child marriage: Comparative evidence from India and Zambia during the COVID-19 pandemic

Pascal Mohamed Mounchid[1], Shruti Shukla[2], Janina Isabel Steinert[2]*

1 Department of Politics and Public Administration, University of Konstanz, Konstanz, Germany, 2 TUM School of Social Sciences and Technology, Technical University of Munich, Germany

* janina.steinert@tum.de

## Abstract

Child marriage is a critical issue in many low- and middle-income countries (LMICs), further exacerbated by the COVID-19 pandemic due to economic instability. This study examines the impact of the pandemic on child marriage rates in India and Zambia, focusing specifically on socio-cultural and economic influences. We collected quantitative and qualitative data from adolescent girls aged 13–18 years in India ($n = 3,049$) and 15–19 years in Zambia ($n = 1,615$) between February and September 2022. Multi-variable linear probability regression analyses were applied to assess the pandemic's effect on child marriage and how it was affected by cultural marriage practices of bride price (Zambia) and dowry (India). While we found no significant increase related to pandemic-induced economic strains in Zambia, we observed a significant increase in child marriage rates related to the pandemic circumstances in India. In-depth analysis of qualitative data indicated that reduced dowry demands and lower wedding costs, resulting from restrictions on guest numbers, drove families to prepone weddings and marry off their daughters during the pandemic. Our findings suggest that economic or health shocks affect child marriage practices differently based on local socio-cultural context.

## Introduction

Child marriage is a global problem rooted in gender inequalities and the belief that the social status of girls and women is inferior to their male counterparts [1]. Child marriage is defined as "any formal marriage or informal union between a child under 18 and an adult or another child" [2]. In a few studies, the term "early marriage" has also been used, with the marriage age of women being defined as under 18 and that of men as under 21 [3]; however, in the current paper, we use the term "child marriage" throughout. Given the detrimental impacts of child marriage on the health and

**Data availability statement:** All data and code is available via https://osf.io/2xgpb/.

**Funding:** Janina Steinert reports financial support for the study was provided by the German Research Foundation (Grant Number: 677875) and World Bicycle Relief. The funders had no role in study design, data collection and analysis, decision to publish, or preparation of the manuscript.

**Competing interests:** The authors have declared that no competing interests exist.

wellbeing of affected children, the United Nations Sustainable Development Goal 5.3 aims to eradicate child marriage by 2030.

Girls are six times more likely than boys to be married before the age of 18 [4]. UNICEF estimates that approximately one in five girls worldwide are married during childhood [2]. The highest rates of child marriage are found in Sub-Saharan Africa, with 34 percent, and in South Asia, with 28 percent of young women married in childhood [5]. For these girls, marriage often marks the abrupt discontinuation of their education, leading to long-time economic dependence on their spouses, an increased risk of facing partner abuse, and a higher likelihood of adolescent pregnancy, despite the associated health risks for both the mother and child [6–10]. While the overall trend of child marriages is declining, scholars have raised concerns about a possible surge in child marriages due to the COVID-19 pandemic [11–13]. Specifically, a Lancet publication from 2020 has warned that the COVID-19 pandemic may put an additional 2.5 million girls around the world at risk of child marriage within the next five years [14].

The circumstances leading to child marriages are complex, involving a blend of more static factors, such as a family's socio-demographic background and the social and patriarchal norms they hold, and dynamic factors, such as economic shocks and unforeseen emergencies like a pandemic [15]. For example, the bride's family might aim to marry off their daughter as soon as she reaches puberty, both for moral reasons such as chastity or family honour and to protect her from potential sexual assault and harassment by other men, which could significantly reduce her marriage prospects and harm the family's honour or social status in the long term [3,16]. Further, marriage decisions can be driven by economic considerations, particularly when families face financial shortages and economic pressure. In patrilocal contexts, where it is customary for the bride to move in with her family-in-law after marriage, marrying off a daughter can reduce the economic burden on her natal household, as scarce financial resources and food no longer need to be shared with one additional household member. In many such settings, this incentive may coexist with dowry institutions, whereby the bride's family transfers assets or payments to the groom's family at marriage [17,18]. Hence, in such systems, the economic benefits of reducing household size by marrying off a daughter must be weighed against the costs associated with providing a dowry. Another economic motivation for marrying off a daughter may arise in bride price systems, where marriage involves a transfer from the groom's family to the bride's family, a practice common in many Sub-Saharan African contexts. Under economic hardship, bride price payments can function as a consumption smoothing strategy, allowing households to alleviate immediate financial constraints. Qualitative evidence from Zambia suggests that some parents and guardians view child marriage as an opportunity to benefit financially from bride price payments to ease household resource constraints and to support younger children [19]. Likewise, empirical evidence from Tanzania shows that marriage payments constitute a major driver of child marriage [20], while more recent work provides broader empirical support for this relationship across countries in Africa, South Asia and the Middle East [15,21,22].

Previous studies have extensively analysed how external economic shocks, particularly those arising from natural disasters, affect child marriage rates [7,18,23]. However, far less is known about how country-specific marriage customs and gender norms moderate these effects. Some direct evidence on marriage responses to economic distress following natural disasters is provided by Chort et al. [24], who examine the interaction between income shocks and bride price norms in Turkey. Exploiting rainfall shocks as an exogenous source of income variation, they show that girls exposed to negative income shocks during adolescence face a significantly higher probability of being married before age 15 in provinces where bride price is prevalent. These findings indicate that child marriage may function as a household coping strategy in response to economic shocks, even outside the poorest country contexts. However, as Haq et al. [25] [p. 9] caution, "mixed studies combining qualitative and quantitative methods are needed to examine the effects of the pandemic on marriage," underscoring the need for integrative approaches that jointly consider the economic shock induced by the COVID-19 pandemic, marriage institutions, and gender norms.

Building on this, our paper examines the links between pandemic-induced household income shocks and the probability of child marriage. We further investigate whether this association is moderated by cultural practices around marriage transfers and differs between bride price and dowry contexts. Specifically, developing a conceptual framework of child marriage, which integrates factors on the individual, socio-demographic, and institutional level, we hypothesise that the pandemic would reduce the likelihood of child marriage in dowry-based societies, because brides' parents may face constraints in mobilising the resources required for dowry payments during periods of financial stress. Conversely, we hypothesise that the pandemic will increase the likelihood of child marriage in bride-price-based systems, as the brides' parents may use transfers from the groom's family as a means of consumption smoothing. Our study presents the first comparative analysis of India and Zambia. It combines quantitative regression methods with qualitative interviews to provide a comprehensive assessment of how the pandemic affected child marriage decisions. These two countries were selected because they represent distinct marriage systems. India is a dowry-based society, while Zambia is characterised by the practice of bride price. At the same time, both countries are located in regions with some of the highest child marriage rates worldwide. Our findings offer crucial insights into how different cultural and economic contexts influence the risk of child marriage, informing policymakers and stakeholders about specific vulnerabilities and mechanisms at play.

The paper proceeds as follows. The background introduces the literature on the factors that influence child marriage and presents the theoretical framework on how the COVID-19 pandemic and cultural factors may have affected child marriage rates in two selected low- and middle-income countries: India and Zambia. In the subsequent section, we describe the study sample, data collection methodology and outline the analysis strategy. The next section presents the descriptive and analytical findings, and the paper concludes with a discussion of wider policy and research implications.

## Theoretical background and related literature

To guide our analysis, we create a conceptual framework as shown in Fig 1. Our framework draws on the model proposed by Kok et al. [26], which identifies four essential levels that contribute to the occurrence of child marriages, namely the (i) individual, (ii) social, (iii) material, and (iv) institutional level [26, p. 8]. At the *individual level*, personal characteristics are considered, including age, educational attainment, employment status, and knowledge of the legal age of marriage for girls and young women. The parents' educational background, household size and the family's status in the community are attributed to the *social level*. The *material level* pertains to the family's economic status, whereas the *institutional level* encompasses the significance of the legal system, societal norms and cultural values and beliefs on gender roles. In our framework, the social and the material level are combined into one *socio-economic level* to streamline the complexity of dimensions while ensuring the inclusion of pertinent micro-mechanisms. Additionally, we extend the framework by Kok et al. [26] with one additional factor: *external shocks*, an additional aspect we introduce to assess the impact of the COVID-19 pandemic on the likelihood of child marriage. Below, we introduce our conceptual framework by drawing

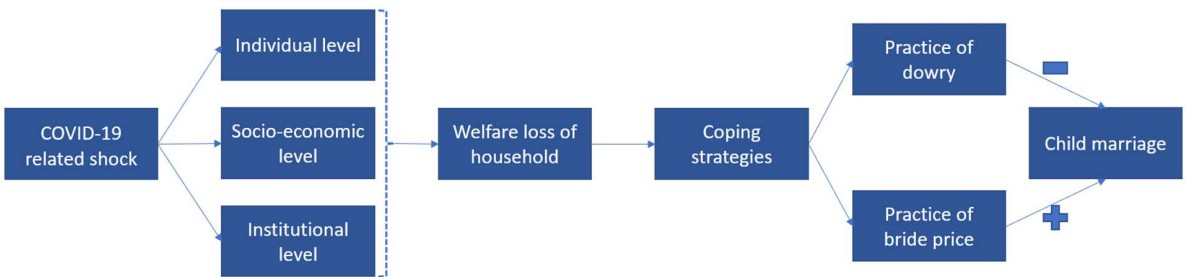

**Fig 1. Conceptual framework of child marriage.** Note: Authors' own compilation.

on evidence on the correlates of child marriage at the individual, socioeconomic, and institutional level as well as links between economic shocks and child marriage. We then delineate our two guiding hypotheses on the differential effects of economic shocks in bride price and dowry systems.

## Correlates of child marriage at the individual and socio-economic level

Concerning the socio-economic conditions that increase the likelihood of child marriage, several studies have identified parents' and the girls' level of education as strong predictors [27–30]. For example, Ali et al. [28] show in their analysis of survey data from Sudan that women who have only acquired primary education are significantly more likely to be married under 18 than those with a higher level of education. It is also known that child marriages particularly often affect illiterate girls [31]. Apart from this, Envulado et al. [32] document a significant positive association between low parental education and increased risk of underage marriage among girls, suggesting that parental educational attainment plays a vital role in shaping marriage-related decisions. At the same time, the authors point out that the economic status of the family is equally important as they observe higher marriage rates among girls whose fathers were farmers, thus relying on lower and more seasonal incomes [32]. In addition to Akter et al. [33] and Hoogeven et al. [34], the study by Kok et al. [26] points to the exceptionally high density of child marriages in regions affected by instability and low income. Corroborating this finding with data from Ethiopia, Aychiluhm et al. [30] show that child marriages take place particularly in rural and impoverished areas of the country. Another study, based in Ethiopia, also explores the links between household wealth and child marriages, revealing that particularly assets owned by a mother are associated with lower rate of girls' marriage rates, whereas paternal assets do not seem to be a key factor [35]. The strong link between economic hardship and child marriages also has important implications for programmatic efforts. Recent evidence from Bangladesh shows that financial incentives can significantly reduce child marriage, whereas empowerment programmes alone may be insufficient. The findings suggest that early marriage may serve as a strategic signal in the marriage market signalling to prospective grooms and their families that a bride is obedient and adheres to conservative gender norms, particularly in contexts dominated by traditional values. In such settings, altering economic incentives appears more effective than attempting to shift gender norms directly [36].

Apart from this, the effect of poverty is particularly strong in rural areas [32]. Al-Ridhwany et al. [29] confirm an urban-rural divide in child marriage rates in their study from northern Iraq. Corroborating this, several studies suggest that the urban–rural divide in child marriage also reflects differences in prevailing gender norms. Using Afrobarometer data from 34 African countries, Charles [37] shows that support for women's equal rights is systematically higher among urban residents and among individuals more exposed to extra-local cultural influences. Similarly, evidence from Demographic and Health Surveys across multiple sub-Saharan African countries indicates that, in most cases, rural residence is associated with a higher likelihood of justifying intimate partner violence against women, a commonly used proxy for conservative gender

norms, although important cross-country variation exists [38]. Another significant social factor contributing to the prevalence of child marriage rates is parental authority. Aychiluhm et al. [30] argue that child marriages are frequently arranged in contexts of limited autonomy of the prospective bride and substantial pressure exerted by her parents.

## Correlates of child marriage at the institutional level

The law around child marriage differs from country to country but often targets the minimum age of marriage and informed consent. To protect girls and boys under the age of 18, countries may adopt various strategies like criminalisation of child marriages, banning marriage below the legally prescribed minimum age, or prescribing a minimum age of marriage without criminalising or banning it [39]. For example, under the Indian Prohibition of Child Marriage Act of 2006, where the minimum age of marriage for girls is 18 and for boys is 21, people involved in performing, conducting, or directing a child marriage can be imprisoned. Further, child marriages can be nullified within two years of the minor reaching adulthood [40]. In Zambia, while the minimum age for marriage is 18 years under the Prohibition of Child Marriage Act, individuals between 16 and 21 were still allowed to get married with parental consent until 2023. In addition, customary laws often took precedence over statutory law, making enforcement of the minimum age difficult [19,41]. Previous studies have shown that legal reforms have a limited impact on child marriage prevalence, and they need to be integrated into broader gender transformative approaches that tackle underlying causes like gender norms, low status of women and girls and cultural values and beliefs [1,42, 43]. Fattah and Camelia [44] used a mixed-methods design in rural Bangladesh and reported high levels of awareness among participants regarding the negative consequences of child marriage. The sample was socio-demographically diverse and included both female and male participants with varying levels of education and self-reported household economic status, ranging from extreme poverty to wealth. The authors further highlight that participants perceived early marriage as a means of protecting family reputation, as it was believed to prevent social stigma in cases where a young girl had experienced sexual assault or harassment.

The belief that the unmarried status of young girls tarnishes the family's reputation following a sexual attack is evident across various regions, including North African countries [21], Ethiopia, Ghana [15,45] and India [7]. Such entrenched value systems appear to be a crucial barrier to the success (or effectiveness) of interventions that seek to enhance the self-determination of young girls [46]. Traditional gender roles thus predetermine the trajectories of girls' lives. In such contexts, the adolescent desire for intimacy, recognition, and social belonging can often be fulfilled only through marriage. Yet this occurs within a normative framework that defines women's roles as "passive, abnegated wives and mothers" [47, p. 5]. The prospect of abstaining from child marriage appears scarcely desirable, as besides the pressure exerted by the social milieu surrounding young girls [48], the entire family would face societal sanctions and ostracism [49,50]. Moreover, religious convictions may also impede the expansion of freedoms for young girls. For instance, Ahonsi et al. [45, p. 2] underscore that "major religious traditions in Ghana encourage early marriage". However, it is essential to note that a universal impact of religious tenets cannot be assumed as two studies by Cislaghi et al. [51,52] from Cameroon elucidate that certain traditions permit marriage only upon reaching the age of maturity.

## The influence of external shocks on child marriage

A body of literature has examined how external shocks, such as those introduced by unforeseen extreme weather events or natural disasters, may affect girls' vulnerability to underage marriage [53]. For example, Ahmed [54] conducted a study investigating the repercussions of the 2010 flood disaster in Pakistan, finding that there was a significant decrease in child marriage rates contrary to the authors' initial expectations. The authors contended that the calamity likely engendered a state of emergency that compelled individuals from forming partnerships as a means of emotional coping. Moreover, Ahmed [54] underscored the possible adherence to legal regulations regarding the minimum age of marriage, notably emphasising the Child Marriage Restraint Act of 1929, which unequivocally prohibits child marriages. However, this

observation's generalisability was challenged by a subsequent study by Ahmed et al. [17], which investigated the impact of a flood disaster in Bangladesh. The researchers highlighted the high poverty rates prevalent in the affected regions and demonstrated an increase in child marriages following the disaster. Drawing on focus group interviews, they argued that parents of married daughters deemed child marriage necessary due to the family's economic constraints stemming from the disaster. Marriage was perceived as a means to transfer the financial responsibility for the daughter to the groom's family, thereby alleviating the financial burden the bride's family faced. Other analyses based on samples from India corroborate this observation [55,56]. Blanco [57] assesses how exposure to different earthquakes within Indonesian provinces between 1994 and 2014 affected girls' chances of being married under age. Specifically, the author reveals a 44% increase in the annual hazard of girls marrying before the age of 18. In line with previous explanations, the author finds that the daughter's marriage functioned as a financial coping strategy through the receipt of a marriage payment. Yet, she also reveals evidence for an additional mechanism, whereby the destruction of school buildings caused a drop in school attendance, consequently increasing marriage rates. Significant increases in child marriage rates following different natural disasters are further shown in a study by Das and Dasgupta [58], who analysed the societal consequences of the 2001 Gujarat earthquake in India, as well as in studies by Tsaneva [59] and Carrico et al. [6] who assess the impact of heat waves in South Asia.

Another type of external shock could have been introduced by the COVID-19 pandemic and the linked economic repercussions caused by lockdown measures and considerable employment losses. The pandemic rapidly exposed the disproportionate burden borne by low- and middle-income countries, many of which entered the crisis with limited fiscal space and institutional capacity. In particular, the ability to cushion the economic consequences of the necessary lockdown measures was barely possible in these countries. Before the pandemic began, there were warnings that poverty rates would rise in the short and medium term, as previous work could no longer be carried out due to the pandemic measures [60]. South Asia, where household income is highly dependent on labour migration, was particularly hard hit by travel and movement restrictions [61]. Household income was also severely reduced in African countries due to the pandemic, and poverty rates rose significantly in a short period [62–64].

Only few empirical studies have evaluated the impacts of the COVID-19 pandemic on child marriage rates. For example, one study by Melnikas et al. [65] draws on data from repeated cross-sectional surveys that were conducted as part of the "More Than Brides Alliance" impact evaluation in 609 villages across four Indian states. Observing child marriage rates over the course of five years, the authors are able to show declines in marriage rates between their baseline and midline measurement, which occurred prior to the COVID-19 pandemic, but then found that this trend stagnated in the course of the pandemic and that child marriage rates increased slightly between midline and endline measurement. Conversely, Batheja et al. [66] leverage data from the Indian National Family Health Survey conducted between 2019–2021 (NFHS-5) and reveal a significant increase in the age at marriage when comparing women who got married after the onset of the pandemic to those married before. The authors suggest that the observed decline in child marriage rates may be driven by a negative income shock, which made it impossible for the bride's parents to afford the wedding and dowry payment for their daughter.

## Research gap

To the best of our knowledge, no study to date has provided a systematic cross-context analysis of child marriage during the pandemic using mixed-method approaches and considering cultural practices in conjunction. Understanding the full impact of the COVID-19 pandemic and external or economic shocks more broadly, requires attention to cultural practices, such as marriage-related transfers, which shape households' strategies for coping with financial strain. Yet, only few studies investigate economic stress by also considering respective local culture context.

For instance, Jibon et al. [67] document pandemic-related economic stress and rising child marriage rates, but without accounting in detail for customs like dowry or bride price. Similarly, studies such as Julianto et al. [68] and a review by Haq et al. [25] highlight economic determinants while neglecting cultural mechanisms. Brandl and Colleran [69] provide a

theoretical link between marriage payments and marriage patterns; however, their analysis remains largely conceptual, as it does not provide quantitative evidence to support the proposed framework [69]. Our paper builds on their contributions by offering comparative evidence from India and Zambia. These two contexts are characterised by contrasting marriage institutions, namely dowry and bride price, which allows us to examine how these mechanisms interact. We provide both statistical estimates and in-depth contextual understanding to uncover the micro mechanisms that drive child marriage under pandemic related stress, offering a more comprehensive perspective than previous research. The subsequent section introduces our theoretical framework and outlines the hypotheses guiding our analysis.

## COVID-19, marriage customs, and child marriage: Theoretical framework and predictions

To examine how income changes such as those caused by natural disasters or external shocks affect marriage prospects, Corno et al. [7] introduced an analytical model, that will guide the hypotheses we test in this paper. In their model, the researchers make three fundamental assumptions about child marriage. First, it is assumed that girls contribute less to the household welfare than they consume, whereas this effect is not prevalent in boys, who are considered as better able to work. Second, it is assumed that, apart from a financial transaction between the groom's and bride's family that is arranged in the context of the marriage, the bride must move-in with the groom's family to shift the financial situation of both households. Third, the model assumes that there is gender balance in the community, leading to a balance in the marriage market. Based on these three assumptions, the micro-mechanisms of the two effect directions are described below, and hypotheses are formulated. To test the predictions of their model, the authors combined weather data with data from the Demographic and Health Survey (DHS) from Sub-Saharan Africa and India. Their analyses revealed that while periods of drought in African countries significantly increased the probability of child marriages by up to three percentage points on average, a similarly significant but *opposite* effect was found in the sample from India, that is, a decrease in the probability of child marriages. Corno et al. [7] argued that the heterogeneous findings that they observe between the two regional contexts are driven by a third factor: customary marriage transfers.

### The practice of bride price

Based on these findings, our first hypothesis is focused on the customary marriage transfer in Zambia: the "bride price". In this form of marriage transfer, it is assumed that the bride's family must receive financial compensation as the relocation of their daughter to the household of the groom's family implies that she can produce economic value for her family-in-law in terms of household or paid labor [7]. Thus, a payment in the form of a bride price is made by the groom's to the bride's family. This payment can be particularly high if the bride is considered young and sexually inexperienced at the time of marriage [70]. If a household's income falls sharply and unexpectedly due to an external shock like a natural disaster, families may have to resort to alternative income generation opportunities, particularly in contexts where poverty rates and economic pressures are already high [71]. Asadullah et al. [55, p. 964] argue that receiving a bride price can serve as a "consumption-smoothing strategy" for families under economic pressure [6,17,57]. Thus, in societies where the practice of bride price is common, child marriage rates can be expected to increase in times of economic shocks. We believe that the same mechanism likely holds true in the context of the COVID-19 pandemic and its linked economic pressures and will therefore test the following hypothesis:

- **Hypothesis 1: In the context of Zambia where the "bride price" custom is prevalent and where families are economically burdened by the COVID-19 pandemic, an increase in child marriages can be expected.**

### The practice of dowry

Our second hypothesis shifts focus to the customary marriage transfer prevalent in India: the "dowry". A dowry denotes a transfer from the bride's family to the groom's family. Dowry payments are often customary in contexts where female

employment rates are generally low and where the daughter-in-law is perceived as placing more of an economic burden on the household she lives in [30]. In India, dowry payments may even amount up to several times the annual income of the bride's family [72,73]. If a household is put under severe economic strain due to an external shock, parents of potential brides are incentivised to suspend or delay marriage arrangements to avoid further financial strain. This prediction is supported by empirical evidence from a panel study conducted in Pakistan: The authors show that in villages where marriage costs are high, early marriage was delayed after households had experienced financial shocks and were therefore more credit constrained [74]. Applying this to the context of the COVID-19 pandemic, we therefore test the following hypothesis:

- **Hypothesis 2: In the context of India where the "dowry" custom is prevalent and where families are economically burdened by the COVID-19 pandemic, a decrease in child marriages can be expected.**

## Methodology

### Study setting

The study focuses on two countries: India and Zambia. Menon [73] estimates that 1.5 million girls in India are married before reaching adulthood, and that almost one in three girls are married before their 18th birthday in Zambia. These high prevalence rates make both countries relevant contexts for studying child marriage. Although the data presented in the following tables may suggest a gradual decline in child marriage rates, the absolute numbers remain alarmingly high [75]. Nevertheless as shown in Table 1, the overall child marriage decreased over the last 25 years, but remains high, meaning that 23.3% of all children were married under age in 2021.

For Zambia, the currently available data indicate a downward trend in child marriage rates over the past decades. As shown in Table 2, the rate has decreased from 46.9% in 1993 to 29% in 2018.

While this long-term progress in both countries is encouraging, there is currently a lack of comprehensive nationally representative data that would allow for a fine-grained assessment of the impact of the COVID-19 pandemic on child marriage rates, which may have significantly disrupted this positive development. According to Equality Now [78], the pandemic has exacerbated many of the underlying drivers of child marriage across Africa—such as poverty, school closures, and reduced access to protection services, which could drive surges in underage marriages.

In India, data were collected in the second most populous state of the country, Maharashtra. Specifically, we targeted peri-urban and urban settlements around the city of Pune as well as rural villages around the city of Sangli. In Maharashtra, approximately 25% of the population lives in poverty, with higher rates in rural areas [79,80]. In Zambia, we collected data in three districts of Zambia's Southern Province: Kalomo, Mazabuka, and Monze. This province is characterised by

**Table 1. Child marriage rates in India by year.**

| 1996 | 2001 | 2006 | 2011 | 2016 | 2021 |
|---|---|---|---|---|---|
| 53.2% | 48.8% | 45.7% | 38.0% | 29.5% | 23.3% |

Data source: Child Marriage Data Portal [76].

**Table 2. Child marriage rates in Zambia by year.**

| 1993 | 1998 | 2003 | 2008 | 2013 | 2018 |
|---|---|---|---|---|---|
| 46.9% | 42.7% | 42.8% | 38.2% | 32.4% | 29.0% |

Data source: Child Marriage Data Portal [77].

low population density, many families living in rural settlements, and a high poverty rate, with 51% of households having lived in extreme poverty in 2022 [19].

## Sampling strategy

**India.** In India, we recruited adolescent girls aged 13–18 years. We started with mapping schools and surrounding communities in both study locations, namely Pune and Sangli. School-attending girls were recruited directly within schools where female and male headmasters and class teachers provided a list of all girls aged 13–18 years. Additionally, to recruit girls who had already dropped out of school, the local team conducted door-to-door visits in the surrounding communities to identify girls within the eligible age range who were no longer attending school. This recruitment process was further facilitated by local community leaders and Anganwadi workers using a random walk procedure, where data collectors began at a randomly selected starting point within each community and then approached every third household, defined as selecting one household after systematically skipping the next two households, following a predefined route until the target sample size was reached. Our final sample in India consisted of 3049 girls, of which 1552 had been recruited from urban and peri-urban settlements in Pune and 1497 from rural villages in Sangli.

**Zambia.** In Zambia, we recruited adolescent girls aged between 15 and 19 years. To identify eligible girls, we approached 100 schools across the three districts of Kalomo, Mazabuka, and Monze. School administrators provided information on the schools' catchment areas and the surrounding communities from which enrolled students typically come. From these, we randomly selected four catchment areas to which enumerator teams were then deployed to recruit age-eligible girls through a random walk procedure, following a similar logic as in India (see above). Our final sample consisted of 1615 girls who were recruited within 92 school clusters. While we had initially planned to cover 100 school clusters, data collection took longer and was more costly than expected since participants' houses were quite spread out, thus requiring long travel distances from one community to the next; data collection was terminated prior to reaching the original target sample size of 2,000 due to budgetary constraints. The final sample included both girls still attending the school as well as girls who had dropped out in the course of the pandemic.

## Data collection

We collected quantitative data via standardised questionnaires administered on tablets with the assistance of female enumerators. Enumerators in India were recruited via two partner NGOs (the SNEH Foundation in Pune and the Astitva Foundation in Sangli) and were fluent in Marathi and Hindi. Data collection was conducted from 1–30 March 2022. In Zambia, enumerators were recruited by the survey firm IPSOS and were fluent in English and Tonga. Data collection took place between 9 May and 9 December 2022. In both countries, enumerators were extensively trained in interview techniques for collecting sensitive data and research ethics, including informed consent procedures, confidentiality, and referral of at-risk participants. Each interview lasted between 60 and 90 minutes and was conducted at participants' homes, on school grounds after lessons (for girls still attending school) or in a convenient community location (such as a community hall), ensuring that confidentiality could be guaranteed in the setting. Similar to previous studies conducted with adolescents in low-literacy populations [81], the surveys were designed as audio- and computer-assisted self-interviews (ACASI) with the intention of maximising confidentiality as well as reducing possible social desirability bias and under-reporting. In both countries, we sought written (or fingerprinted, in the case of limited literacy) informed consent from girls who were 18 years or above. For girls under the age of 18 years (the majority of our sample), we adopted a two-stage consent procedure by first collecting written informed consent from a parent or legal guardian and subsequently collecting verbal assent from each girl.

In addition to quantitative data, we collected qualitative data. In India, we collected extensive qualitative data from a total of eight focus group discussions (FGDs) and nine individual in-depth interviews. The focus group discussions were carried out prior to the quantitative data collection with groups of mothers (2), fathers (2), adolescent girls (2), and social workers

(2). In-depth interviews were conducted after completion of the quantitative survey, and participants were sampled purposely based on having disclosed any of the following adverse outcomes: (i) having experienced family or intimate partner violence in the previous year, (ii) having been married underage, and (iii) reporting a current or past pregnancy. Interview guides were aimed at eliciting the harmful impacts of the COVID-19 pandemic and at discussing their underlying causes. In Zambia, due to budget constraints, we could only conduct a total of three FGDs. These were carried out prior to the quantitative data collection with one group of mothers and two groups of adolescent girls. Similar to the FGDs in India, discussion guides were focused on the harmful impacts of the COVID-19 pandemic. All discussions and interviews were conducted in the respective local language (i.e., Marathi, Hindi or Tonga) and recorded verbatim and then transcribed and translated into English.

## Measures

**Dependent variable.** The dependent variable of interest is the girls' marital status, which was dummy-coded as 1 if the girl reported being currently married, divorced or widowed and if the wedding ceremony had occurred before the girl had reached her 18th birthday. The survey asked girls to indicate the date of their wedding ceremony, which was then cross-checked against their age information to ensure that only marriages that had occurred before their 18th birthday were included.

**Explanatory variables.** We included explanatory variables reflecting the different levels introduced by Kok et al.'s [26] theoretical framework outlined above. On the individual level, we included (i) girls' age and (ii) caste (for the Indian sample only). We decided not to include girls' educational status due to endogeneity concerns, given that school dropout could likely be a consequence of child marriage rather than a predictor (see, for example, Delprato et al. [82]) and that our cross-sectional data did not allow us to differentiate between cause and consequence. On the socio-economic level, we included the (i) education status of both the mother and father, defined as a binary variable that was coded as 1 if the mother/father reported having completed secondary school; and (ii) households' wealth status, which was an additive index capturing the amount of assets owned by a household as well as achieved living standard indicators such as electricity, flush toilet, and drinking water in the house (see S1 Table for a list of individual indicators of included aggregate measures). On the institutional level, we sought to measure gender norms by including an additive index that measured girls' approval of violence against women, whereby a higher index score denoted a higher acceptance of violence. Specifically, respondents had to state whether they agreed or disagreed with seven different statements asking about scenarios in which it would be justifiable for a husband to beat his wife, for example, if the wife *"goes out without telling her husband"* or *"hides money from her husband"*. Lastly, we included our core explanatory variable, capturing the economic shock dimension, which is defined as the socioeconomic impact of the COVID-19 pandemic. The variable was an additive index composed of six individual indicators asking whether the girls' family had (i) lost employment or their source of livelihood, (ii) reduced work hours, (iii) faced financial shortages, (iv) gone hungry, (v) faced a higher care burden, and (vi) accommodated returning work migrants at home in consequence of the pandemic and its linked lockdown restrictions (see S1 Table). Higher index scores thereby indicate greater economic pressure induced by the pandemic.

## Analysis

Associations between explanatory variables on the individual, socio-economic, institutional, and external shock levels with the likelihood of child marriage were estimated based on multi-variable linear probability regression analyses that were run separately for the respective samples from India and Zambia. We chose linear probability regression over logistic regression to facilitate the interpretation of coefficients, which indicate percentage-point changes in the likelihood of being exposed to child marriage for each included explanatory variable. For ease of interpretation, we also estimated a logistic regression model to calculate the predicted probability of child marriage for a low- and high-intensity economic shock scenario, with the latter being defined as a household having experienced at least two of the pandemic's economic repercussions listed in S1 Table. The sampled age ranges are narrow (13–18 in India; 15–19 in Zambia), limiting the potential for

meaningful non-linear effects; within these age bands, the likelihood of child marriage tends to increase linearly with age, and including an age-squared term would add unnecessary complexity. Analyses were carried out in R, Version 4.3.3. All data and code is available via https://osf.io/2xgpb/.

### Ethics

In India, the study was approved by the ethics committee of the Indian Institute of Technology Gandhinagar (IITG) in November 2021. In Zambia, the study was approved by the National Health Research Authority in July 2022. Both study components were additionally approved by the ethics committee of the medical faculty at the Technical University of Munich (TUM) on 29 December 2021. Additional information regarding the ethical, cultural, and scientific considerations specific to inclusivity in global research is included in the Supporting Information (S1 Checklist in S1 File).

## Results

### Socioeconomic characteristics, child marriage, and impact of the pandemic

Descriptive information on girls' socio-demographic status, exposure to child marriage, and COVID-19 impacts are summarised in Table 3, separately for the sample from India and Zambia.

**India.** The average age of girls in the Indian sample was 15.6 years and more than two thirds of girls belonged to a lower caste. 16.5% of girls reported that they engaged in some form of income generation, and their households owned, on average, 6.7 out of a list of 14 assets. 28% of girls' mothers, compared to 44% of girls' fathers had completed secondary school, thus reflecting stark gender disparities in parents' education status.

In our sample, 9% of adolescent girls in India were married underage. Brides' average age at marriage was 15.4 years, whereas the rate of being married increased with girls' age. Grooms were generally older than brides, with an average age of 22 years, thus also marking considerable age gaps between spouses. To illustrate this, more than 40% of the girls in our Indian sample aged 18 years were already married (see S1 Fig). The youngest bride in our sample was 13 years old when the wedding ceremony took place. Of the married girls, only 21% indicated that they got married because of love, whereas 71% indicated that their marriage was arranged by their family. Further, 9% of girls disclosed that they perceived their marriage as a forced marriage. A dowry was paid in 24% of the recorded marriages in the sample. We also captured girls' attitude towards child marriage by using a hypothetical scenario in which we asked them how they would feel if their sister or a close female friend were to get married before the age of 18. Notably, 88% of the girls in our Indian sample disclosed that they would feel that *"she is too young to get married"* (see S2 Table).

In India, the economic impact that the COVID-19 pandemic on the families and households of girls was substantial. Specifically, 60% of girls reported that at least one household member had lost their employment during the pandemic, and 55% reported productivity losses. In addition, 84% of households experienced financial shortages due to the pandemic and 35% of households struggled to provide enough food for everyone in the family. In 56% of households, the burden of unpaid work increased as a consequence of the pandemic, and in 10% of households, returning work migrants had to be accommodated as additional household members in the already strained family.

**Zambia.** The average age of girls in the Zambian sample was 16.7 years. 43% of these girls were engaged in some form of income generation activity, and 30% lived in polygamous households, which could partly explain the generally bigger household size in the Zambian sample, relative to the Indian sample (7.8 household members versus 4.1 household members, respectively). Overall household wealth in Zambia was also lower than in India, with an average of 4.7 (compared to 6.7 in India) out of 14 assets owned and living standard indicators achieved by the family. 34% of girls' mothers and 39% of girls' fathers had completed secondary education.

In Zambia, 7% of adolescent girls in our sample were married before they had reached age 18. The average marriage age was higher than in India, at 17.1 years (see S2 Fig). The youngest bride in the Zambian sample was 15 years old. Of those married, 59% reported that they got married because of love—a larger share of married girls than in the Indian

**Table 3. Sociodemographic and COVID-19 impact data from India and Zambia.**

| Variable | India | | Zambia | |
|---|---|---|---|---|
| | Mean (SD) | n | Mean (SD) | n |
| *Sociodemographic Information* | | | | |
| Age | 15.558 (1.482) | 3049 | 16.716 (1.415) | 1615 |
| Lower/Scheduled Caste | 0.669 (0.471) | 3049 | | |
| Income generating activity | 0.165 (0.371) | 3049 | 0.043 (0.205) | 1615 |
| Number of assets/living standards indicators | 6.700 (2.044) | 3049 | 4.661 (2.111) | 1615 |
| Number of household members | 4.082 (1.602) | 3049 | 7.780 (4.146) | 1651 |
| Polygamous household | | | 0.291 (0.454) | 1615 |
| Mother secondary education | 0.284 (0.451) | 3049 | 0.343 (0.475) | 1615 |
| Father secondary education | 0.442 (0.496) | 3049 | 0.387 (0.487) | 1615 |
| *Child Marriage Indicators* | | | | |
| Child Marriage | 0.091 (0.288) | 3049 | 0.059 (0.236) | 1615 |
| Age at marriage bride | 15.411 (1.876) | 286 | 17.068 (1.039) | 119 |
| Age at marriage husband | 22.035 (3.228) | 286 | 20.687 (4.018) | 119 |
| Dowry/bride price paid | 0.238 (0.426) | 286 | 0.658 (0.476) | 119 |
| Love marriage | 0.207 (0.405) | 286 | 0.589 (0.493) | 119 |
| Arranged marriage (by parents) | 0.705 (0.456) | 286 | 0.376 (0.486) | 119 |
| Forced marriage | 0.087 (0.283) | 286 | 0.034 (0.182) | 119 |
| *COVID-19 Impact* | | | | |
| Loss of employment during COVID | 0.595 (0.490) | 3094 | 0.064 (0.244) | 1615 |
| Reduced hours for income generation during COVID | 0.551 (0.496) | 3094 | 0.124 (0.330) | 1615 |
| Food shortage during COVID | 0.351 (0.477) | 3094 | 0.190 (0.392) | 1615 |
| More unpaid care work during COVID | 0.559 (0.496) | 3094 | 0.126 (0.332) | 1615 |
| Returning migrants in home during COVID | 0.101 (0.301) | 3094 | 0.111 (0.315) | 1615 |
| Financial shortages during COVID | 0.841 (0.365) | 3094 | 0.390 (0.487) | 1615 |

Notes: Age at marriage, type of marriage, and financial transfers at marriage are only reported for the subsamples of married girls, namely 286 girls in India and 119 girls in Zambia.

sample. Further reflecting this, only 3% of girls indicated that they were forced to get married by family members. In 66% of all reported marriages, a bride price was paid. Child marriage as a norm and practice was rejected more strongly among girls in Zambia as compared to girls in India. 98% of the Zambian girls in our sample claimed that they would feel that their sister or friend would be too young if she got married before reaching 18 years of age (see S2 Table).

Lastly, the detrimental socioeconomic impacts of the COVID-19 pandemic were also reflected in the sample from Zambia, albeit less strongly than in India. Specifically, 6% of girls in Zambia reported that at least one household member had lost their employment during the pandemic, and 12% reported that someone had faced productivity and income losses. 84% of Zambian households faced financial shortages caused by the pandemic, and 19% of respondents reported that they or someone in their family had to go without eating for a whole day because of lack of money and resources during the pandemic. The unpaid care burden increased in 13% of households, and 11% took in work migrants who came back to their rural villages during the pandemic.

## Correlates of child marriage

We explored possible factors associated with child marriage on the (i) individual, (ii) socio-economic, (iii) institutional, and (iv) economic shock level based on separate linear probability regression models (see Fig 2).

**Zambia**

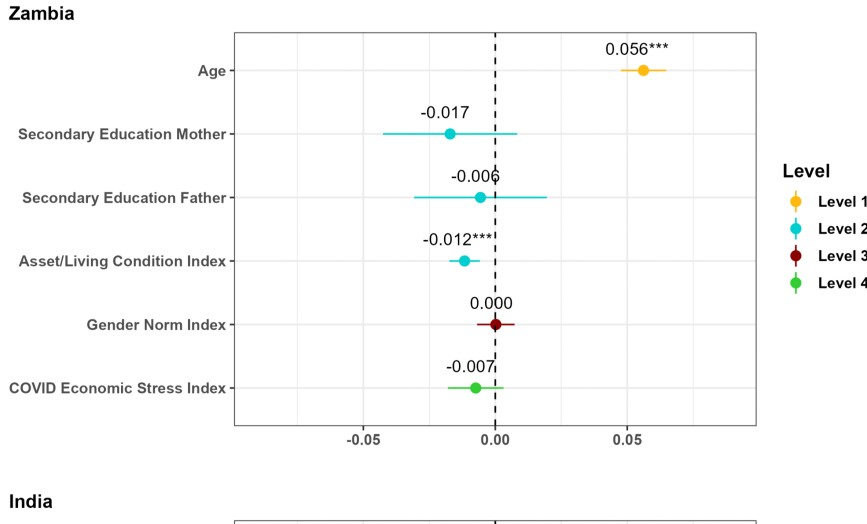

**India**

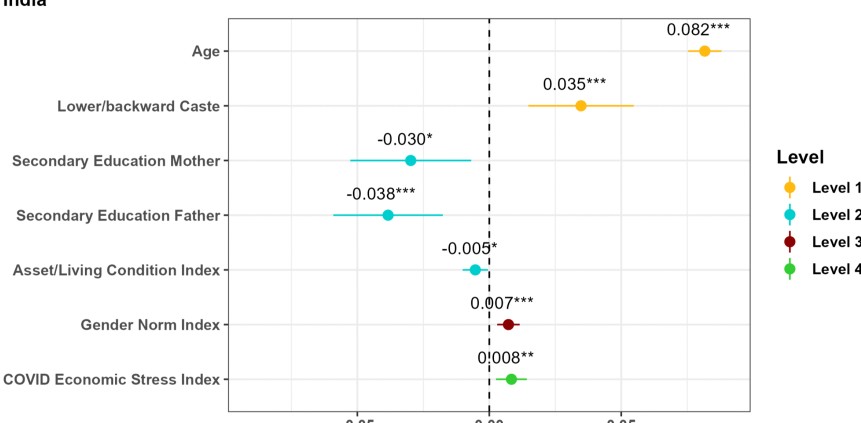

**Fig 2. Correlates of child marriage on different levels.** Note: Coefficients from linear probability regression model. *p<0.05, **p<0.01, ***p<0.001. Level 1 (yellow): individual level, Level 2 (blue): socio-economic level, Level 3 (red): institutional level, Level 4 (green): economic shock level. Coefficients at different levels of the theoretical framework were estimated in separate models to avoid multicollinearity and endogeneity concerns. In each model, we controlled for individual-level factors, namely age and caste in India.

In the sample from Zambia, only two factors were significantly associated with the outcome variable. Specifically, the strongest predictor of girls' marital status was their age, with a six percentage point increase in the probability of being married with every additional year of life. In addition, the rate of child marriages was significantly lower in households with higher living standards and wealth. The effect translates into a significant one percentage point decrease in the probability that a girl is married under age with every additional asset or living standard indicator that a household acquires. Notably, we did not observe any significant associations between child marriage and parents' education status or girls' internalised gender norms. Importantly, contrary to the predictions made by our *Hypothesis 1*, we did not observe a significant increase in child marriage rates among households that were more adversely affected by the economic pressures of the COVID-19 pandemic. In fact, the coefficient for the COVID Economic Stress Index even points in the opposite direction, albeit not being significant. Thus, we must reject *Hypothesis 1*, as we find no indication that the economic shock induced by the COVID-19 pandemic has prompted households to seek alternative income sources through a bride price and marrying off their daughters as a consumption-smoothing strategy.

In the sample from India, girls' age was also a significant correlate of child marriage, translating into an eight percent-age point increase in girls' likelihood of being married with every year of age. Girls from lower castes were also more at risk of being married under age, whereby their probability of being married was 3.5 percentage points higher as compared to girls who were from higher castes. On the socio-economic level, girls whose fathers and/or mothers had completed secondary schooling were significantly less likely to become child brides, thus suggesting that parents' higher education is a protective factor with respect to child marriages. Similar to what we observed in Zambia, girls from economically better-off households were also less likely to be married under age, whereby any additional asset or living standard indica-tor translated into a 0.5 percentage-point decrease in the probability of child marriage. In addition, our analyses revealed a significant link between child marriage and the institutional-level factor of gender norms. Specifically, girls who were more approving of violence against women (reflecting stronger internalised patriarchal gender norms) were significantly more likely to be married before reaching the age of 18, with every additional scenario (out of 7) in which a girl considers it justifiable for a husband to beat his wife translating into a 0.7 percentage point increase in the probability of being married. Finally, our analysis fails to support the predictions made by our *Hypothesis 2*. While—based on our theoretical frame-work—we expected a decrease in child marriage rates within a dowry-based system like India, our regression reveals the opposite trend. Specifically, we observe a significant 0.8 percentage-point increase in the likelihood of girls being married underage with each additional economic pressure faced by a household due to the COVID-19 pandemic.

Lastly, for a more nuanced interpretation of the economic shock effect, we calculate the predicted probability of child marriage for a low- and high-intensity economic harm scenario. As shown in Fig 3, in Zambia, we do not observe any significant differences in the predicted probability of child marriage when comparing a hypothetical household that has experienced no substantial economic shock due to the pandemic with a hypothetical household that has suffered significant economic harm, for example through income losses or food shortages. Notably, the predicted probability even drops from 6.30 percent to 4.02 percent with higher economic pressures, thus suggesting that families in Zambia were not resorting to child marriages with the intention to smooth consumption by means of a bride price. Conversely, in India, we report a significant increase in the predicted probability of child marriages: While in the low-intensity eco-nomic harm scenario, there is only a 6.50 percent chance that a girl is married under age, this risk almost doubles and

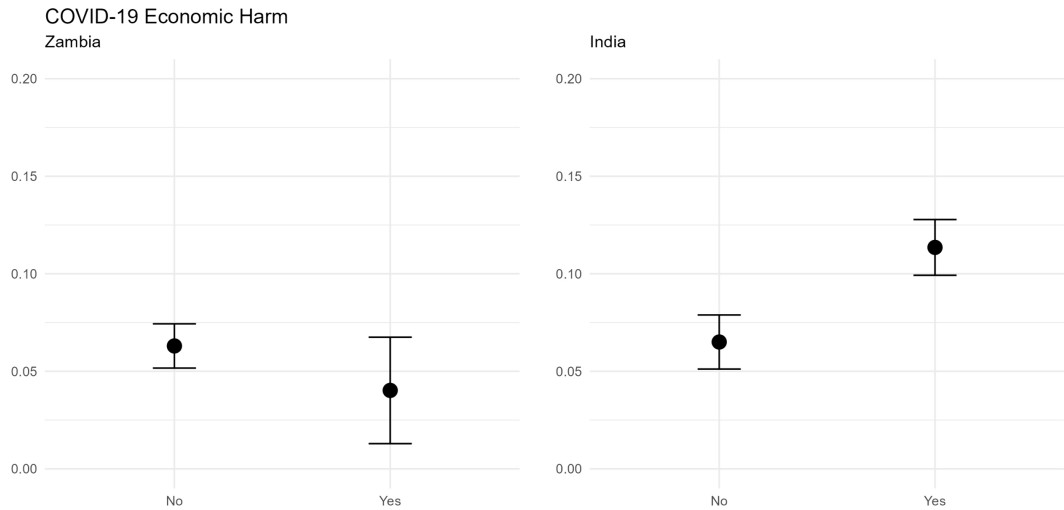

**Fig 3. Predicted probability of child marriage for different economic harm scenarios.** Predicted probabilities with robust standard errors based on logistic regression model, controlling for age and caste (in India). COVID-19 Economic Harm was coded as 1 ("yes") if a household had suffered two or more economic repercussions (see S1 Table), and as 0 ("no") otherwise.

increases to a 11.35 percent probability for girls to become child brides in the high-intensity economic harm scenario. The significant surge in the projected likelihood of child marriages in India implies that dowry payments did not function as a hindrance to the establishment of formal unions in face of the economic strains experienced during the COVID-19 pandemic.

**Qualitative findings on the associations between COVID-19 economic pressures, marriage customs, and child marriage**

**Zambia: Bride price system.** Our analysis in Zambia, a society that still holds bride price traditions, did not reveal an elevated risk of child marriage in the context of the COVID-19 pandemic, thereby refuting our *Hypothesis 1*. While our quantitative data could not confirm that the economic stress induced by the COVID-19 pandemic increased early marriage rates, the qualitative data from our focus group discussions can offer more nuanced insights into the potential factors driving still occurring child marriages during this period. Notably, one significant contributing factor might have been the school closures implemented due to the pandemic. These closures resulted in increased free time, allowing boys and girls to spend more time together and form relationships. As one adolescent girl described:

*"Some days back, before Covid-19, we all [adolescent girls] used to go to school. There was no time to meet boys. But when Covid-19 started, schools started closing and that is when people had the chance to say 'ah! let us get married'. When schools closed, they [girls and boys] started marrying each other while they were still young."*

Additionally, the prolonged closure of schools may have diminished girls' educational aspirations, shifting the trade-off between pursuing more years of schooling versus opting for early marriage in favour of the latter. Accordingly, a girl stated during the discussion:

*"Even the thoughts of being in class had finished. They [friends from school] said they did not know what they were going to write, when they would go to class again, so they just thought of getting married. All the thoughts about school had vanished, there were only thoughts of playing that remained."*

Lastly, the economic pressures induced by the pandemic may have made it more difficult for parents to afford their children's and particularly their daughters' school fees. This financ\ial strain could have pushed these daughters into earlier marriages, as expressed in the following statement by the mother of an adolescent girl in Mazabuka:

*"Sometimes, what causes that [early marriage] is that there is no one to pay for her [adolescent girls'] school fee and she is just living at home. Then she will think of getting married instead of staying at home."*

**India: Dowry system.** Our analysis of data from India, a society where dowry practices are prevalent, did not support *Hypothesis 2*. This hypothesis, aligned with Anderson et al. [83], proposed that child marriage rates would likely decrease during periods of economic pressure. A possible explanation for the increase in child marriages that we observe could be that the groom's family did not demand dowry transfers, acknowledging the financial struggles many families faced during the pandemic. Indeed, only 24% of married girls indicated that a dowry was transferred for their marriage (see Table 1), and in 31% of the wedding ceremonies within our sample, there were reportedly no transfers of gifts such as jewelry, food, or gold. While alleviating financial distress for brides' families, these reduced dowry demands might also have made the pandemic period an opportune time for marrying off daughters without conventional economic obligations. Evidence gathered during focus group discussions and in-depth interviews corroborates this explanation, as captured in the following quotation by an adolescent girl from Sangli:

*"During Corona, everyone faced financial losses and they [families] didn't have money for their daughter's wedding. So if they got a family who was not asking for much [dowry] then they married off the girl and there were no expenses towards the wedding as there would be no guests due to Corona [lockdown]."*

Supporting this, a mother participating in a focus group discussion in Pune confirmed: *"The overall cost of the weddings has reduced a lot… really a lot."*. Similarly, a girls' father from Sangli noted: *"It [dowry] is still there, but it has reduced a lot. Recently, one of our relatives got married, the boy's family didn't take a single rupee."*

Together, these findings indicate that reductions in dowry demands and wedding ceremony costs reflect a single cost-reduction mechanism through which the pandemic altered marriage decisions. In addition to reduced dowry demands, the costs associated with wedding ceremonies emerged as a closely related factor. In India, wedding ceremonies often involve several hundred guests, and it is customary for the bride's family to bear this expense. However, our qualitative data highlighted that during the COVID-19 pandemic and the related contact restrictions, wedding ceremonies were significantly smaller, with only a few guests allowed to attend, and thus substantially cheaper. Corroborating this, girls' parents confirmed during focus group discussions:

*"Earlier [prior to the pandemic] there was the cost of guests, and also the cost for their invitations etc. Now hardly anyone is invited, and people also don't mind not being invited. They understand, that because of COVID, it is difficult to have regular weddings. Earlier, people would literally ask: 'Why didn't you invite us for the wedding?' But now everyone knows and understands. Now even the children who are getting married, don't have such expectations; They're happy even if the ceremony takes place at home with immediate family."*

Thus, the pandemic may have presented an attractive economic opportunity for parents to marry off their daughters at a significantly lower cost than usual. Some of the marriages even seemed to have been arranged hastily and in an ad-hoc fashion to benefit from the low-cost circumstance. Reflecting this, an adolescent girl from Pune described the following situation during an in-depth interview:

*"Some people [adolescents in the participant's village] got married in the evening due to fear of police [during the lockdown] with very few people. They got married in the morning at 3 a.m. or 4 a.m. in the temple with only two three people present for the marriage. Even the girl did not know that she was going to get married."*

Another mother added:

*"Indian weddings always take place with great pomp. There are 2-3 hotels near where we live, and each time we see at least 300-400 cars parked, some distance from the hotels [for the ceremony]. So the expense is definitely extracted from the girls' family. But now [during the pandemic], a lot of the cost can be reduced, by having simple weddings in homes or in temples. My own cousins' brother got married recently, and he managed to save 5–10 lakhs (between 6,000–11,000 EUR) by having a very simple wedding, which we all attended virtually."*

Lastly, by marrying off their daughter, the bride's natal family may benefit from having fewer mouths to feed, as she moves in with her in-laws, thereby reducing the strain on their limited resources. For instance, one girl recounted:

*"For my marriage, my family did not have to carry any expenses as my father had told them my in-laws that he can only give the girl and nothing else since we are poor, but the boy's family had everything and took me."*

Similarly, some participants described how marrying off a daughter could help parents cope with the health-related pressures and expenses of the pandemic. Reflecting this, one father emphasised during a group discussion: *"If there*

*is hospitalisation in the family, then it is good to get her [the daughter] married as it is difficult to continue her education then."*

Taken together, these accounts suggest that the pandemic created a window of opportunity to marry off daughters at substantially reduced cost. This is further supported quantitatively: 19.49% of married girls in India reported that their wedding was preponed as a direct consequence of the pandemic, compared to only 6.03% in Zambia (see S3 Table).

## Discussion

Our study aimed to identify the effect of the COVID-19 pandemic and country-specific cultural factors on the risk of child marriage in India and Zambia.

### Findings and implications for the cultural context of India

In India, 9% of adolescent girls included in the study were married underage, with significant age gaps between spouses. Marriages were predominantly arranged by families, dowry was paid in 24% of the marriages and economic pressures during the pandemic increased the likelihood of child marriage.

Our findings on the individual, socio-economic, and institutional correlates of child marriages integrated into our conceptual framework align with existing literature. Similar to previous studies, we also find that individual-level factors such as higher age and lower caste were associated with an increased risk of child marriage [84,85]. In many LMICs, menarche translates into marriage eligibility, which is deeply rooted in patriarchal norms like the maintenance of purity before marriage and upholding the family reputation [86–88]. As a result, societal and familial pressure to marry off girls may intensify as they grow older. A previous study conducted with the same sample of 3049 adolescent girls in India presented that social expectations around marrying early were one of the main drivers of adolescent marriage and pregnancy [53]. Significant age differences between adolescent girls and their partners, as seen in this sample, can also lead to power imbalances in the relationship, leading to loss of sexual and reproductive health-related decision-making power and even intimate partner violence [89].

The higher incidence of child marriage among lower caste families reveals the compounded disadvantages and social marginalisation faced by these groups. Other studies also corroborate these findings [90,91]. At the socio-economic level, parent's education level and household wealth were negatively associated with the probability of child marriage. Studies from other LMICs have also shown that higher parental education and wealth protect against child marriages [35,46,92]. Additionally, the institutional-level factor of gender norms, here captured by a measure of justifying violence against women, emerged as a critical factor influencing the incidence of child marriage. Previous studies from India and Pakistan have also presented that women married as children were more likely to justify spousal violence [93,94]. Others have highlighted that women who were married as children or married as adolescent girls face increased violence and controlling behaviour from their partners [81,95,96].

Finally, with regards to the impact of economic shocks, our analysis did not support the hypothesis derived from our conceptual framework that child marriage rates would likely decrease in consequence of the economic shock caused by the COVID-19 pandemic. In contrast, we found significantly higher rates of child marriages among families that experienced substantial economic hardships due to the pandemic. Nearly one-fifth of the child brides in our sample reported that the timing of their marriage was influenced by the pandemic, causing their marriages to occur earlier than they would have under normal circumstances. Qualitative interviews with married adolescents highlighted that the lack of dowry demand and low wedding costs made marrying during lockdown lucrative. While most previous studies observe a negative relationship between economic shocks and child marriage rates in dowry-based societies, a study from Bangladesh also presented that in the aftermath of a flood, marriages were less expensive as most households suffered economically and social expectations around dowry and wedding expenses were lower, leading to families marrying off their daughters [17]. Likewise, another study in the same setting from 2020 showed that households coped with economic hardship by marrying off their daughters to reduce household consumption [6].

## Findings and implications for the cultural context of Zambia

In Zambia, the rate of child marriage was slightly lower at 7%, with a higher proportion of marriages driven by romantic relationships rather than family arrangements. Higher household wealth was strongly associated with lower rates of child marriage in both countries. But while the COVID-19-related economic shock played a crucial role in the family's financial situation, it did not lead to an increase in child marriages as predicted by our conceptual framework.

Like in the Indian context, in Zambia as well, inequitable gender norms that prioritise women's roles as wives and household caretakers and men's role as the breadwinner often result in low investment in girls' education and future economic prospects. The drivers of child marriage in this sample were highlighted during interviews with girls. They mentioned that school closure due to the pandemic, economic pressures leading to non-payment of school fees, and prolonged periods of tedium due to the lockdown made families and girls lose interest in future education. Similar channels were also identified in a qualitative study in Indonesia after the pandemic [13]. A combination of inequitable norms and a lack of avenues to pursue education pushes girls to form financially beneficial relationships in return for sex, leading to school dropout, adolescent pregnancies and subsequent child marriages [97–99].

The significant association between household wealth and lower child marriage rates aligns with previous research from LMICs, which emphasises the protective effect of economic stability against early marriage [100–103]. In our sample, despite significant economic hardships that households faced during the COVID-19 pandemic, like loss of employment or income, there was no corresponding increase in child marriage rates, suggesting that Zambian families employed different coping mechanisms than their Indian counterparts. Families in Zambia did not marry their daughters in lieu of receiving the bride price as a consumption smoothing strategy. This finding is in contrast to previous studies that indicated an elevated risk of child marriages in societies with bride price traditions following natural disasters or income shocks [6,18,26,55,56,58,59,83]. One possible explanation for our findings is that financial hardship during the pandemic affected households broadly, including families of prospective grooms. This may have reduced their ability to afford high and attractive bride price payments, thereby lowering the incentives for parents of potential brides to marry off their daughters early during periods of widespread economic distress. While our findings clearly differ from those reported in Corno et al. [7] and from the predictions made by our conceptual framework, it is important to note that the COVID-19 pandemic represents an unprecedented external shock that is not fully comparable to natural disasters. In particular, the prolonged economic consequences of lockdown policies likely imposed a distinct form of financial strain on households, differing from the strain typically observed during environmental or humanitarian disasters, which are often accompanied by substantial inflows of humanitarian aid and relief programmes. Moreover, our results indicated that social distancing policies were deliberately leveraged to reduce expenditures on daughters' weddings, a mechanism specific to the pandemic context and unlikely to generalise to other types of shocks. Against this background, our conceptual framework cannot be validated in the specific context of the COVID-19 pandemic.

## Limitations and avenues for future research

This study has several limitations. The cross-sectional nature of the data does not allow us to draw causal conclusions on how the risk factors are linked to child marriage. To mitigate this, we present a conceptual framework based on previous studies and qualitative interviews with adolescent girls to unpack potential channels linking economic, social, cultural, and disruptive factors to child marriage outcomes. Apart from this, we did not collect any data prior to the pandemic based on which we could estimate the increase in child marriage rates caused by the pandemic. However, we apply an extensive list of measures that represent nuanced variations in the extent to which households were negatively affected by the pandemic. Further, data collection was limited to two rural and semi-urban locations in India and three rural locations in Zambia, which may affect the generalisability of our findings. Lastly, the relatively small number of focus group discussions conducted in Zambia limits the depth of contextual insights that can be drawn for this setting.

Future studies should aim to collect longitudinal data and leverage quasi-experimental designs to establish causal relationships between the highlighted risk factors and child marriage. Additionally, incorporating qualitative interviews with not only adolescent girls but also their parents and relatives could provide deeper insights into the coping mechanisms and cultural contexts that drive child marriage, helping to unpack the nuanced ways in which economic, social, and cultural factors interact to influence marriage decisions.

### Implications

The findings from this study have significant implications for intervention development and policy making aimed at reducing child marriage in LMICs both during public health emergencies (PHEs) and beyond. Restrictive gender norms are a significant barrier for girls in achieving their aspirations and avoiding child marriage. Eroding this would require interventions to promote communication and support from their parents and community. Such programmes could help adolescents and parents to develop ways of coping with conflict and setting shared boundaries. One such intervention is the Families Matter! Programme in Tanzania that aims to empower caregivers to discuss sexuality and practice positive parenting skills with their children [104]. Building on this, future interventions could be adapted to include discussions and activities specifically focused on sexual reproductive health and child marriage.

A recent systematic review of the association between PHEs and various gender outcomes highlighted that PHE-related response strategies like home confinement and closure of schools, along with associated economic losses at both national and household levels are key channels behind the increase in child marriage in LMICs [53]. While these measures are essential to curb new infections, governments and policymakers should focus on complementary measures to safeguard adolescents' health and life outcomes. These include providing safe spaces for them to socialise and receive education, organising mobile youth-friendly health and counselling services, subsidising or removing school fees to allow girls to return to school, and integrating child marriage prevention messages into the broader policy landscape [105–109]. Lastly, addressing structural barriers like poverty, low education, and limited employment opportunities requires continuous development of policies and programmes at a national level to increase the economic resilience of families, especially during crises. This could include social protection programmes, job creation initiatives, and support for small businesses.

### Supporting information

**S1 Table. Construction of index measures.**
(TEX)

**S2 Table. Girls' attitudes towards child marriage.**
(TEX)

**S3 Table. Economic and temporal aspects of marriages.**
(TEX)

**S1 Fig. Child marriage rate by girls' age: India.**
(TIF)

**S2 Fig. Child marriage rate by girls' age: Zambia.**
(TIF)

**S1 File. Supplementary Information.**
(PDF)

## Acknowledgments

For data collected in India, we wish to thank Rucha Vasumati Satish, the SNEH Foundation and ASTITVA Foundation as well as our dedicated team of enumerators: Vandana Murlidhar Kamble, Jayshree Kamble, Fairaza Mulla, Manali Amol Vhankade, Madhuri Malhari Mandale, Kajal Vitthal Garale, Shivani Aapte, Rajashree Nitin Vhankade, Deepa Ramhari Bhusanar, Sunita Gadhire, Sonal Umesh Bhoyar, Priyanka Raju Chandane, Nafisabanu Farid Shaikh, Rashmi Adinath Khandagale, Vimal Hemant Pise, Savita Ramdas Durge, Sunita Salunke, Dorthi Robin Joseph, Aruna Hitendra Kamble, Pallavi Abasaheb Sangale, Jyoti Lahurao Surywanshi, Shubhangi Sachin Salve, Bhagyashree Patole, Shraddha Bhargande, Samina Shaikh, and Shilpa Samant. For data collected in Zambia, we wish to thank the team from IPSOS, including Oscar Mutinda, Tukiya Mbewe, and Winnie Sambu, as well as our other project partners Alisha Myers, Lawrence Banda, and Ana Garcia Hernandez. We are also deeply grateful to our enumerators: Oness Hinamanjolo, Adinator Mukonka, Sarah Mapiki, Nalucha Sikananu, Mable Mukulumwa, Chilwana Chilenga, Mamire Hamiyanda, Brenda Nkolola, Nachamba Chipompwe, Pumulo Njani, Cynthia Choongo, Cynthia Handili, Namwinga Himonga, Martha Kayawe, Latoya Ntambu, Linety Siagwelele, Annie Masheke, Luyando Himonga, Zithe Mwale, and Monde Masheke. We also wish to thank all girls who volunteered their time to participate in this research study and shared their stories with us.

## Author contributions

**Conceptualization:** Shruti Shukla.

**Data curation:** Pascal Mohamed Mounchid, Shruti Shukla, Janina Isabel Steinert.

**Formal analysis:** Pascal Mohamed Mounchid, Janina Isabel Steinert.

**Funding acquisition:** Janina Isabel Steinert.

**Investigation:** Janina Isabel Steinert.

**Methodology:** Pascal Mohamed Mounchid, Shruti Shukla, Janina Isabel Steinert.

**Project administration:** Shruti Shukla, Janina Isabel Steinert.

**Supervision:** Janina Isabel Steinert.

**Validation:** Janina Isabel Steinert.

**Visualization:** Pascal Mohamed Mounchid.

**Writing – original draft:** Pascal Mohamed Mounchid, Janina Isabel Steinert.

**Writing – review & editing:** Pascal Mohamed Mounchid, Shruti Shukla, Janina Isabel Steinert.

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
