## [Decision Letter · Decision Letter 0]

26 Nov 2025

PONE-D-25-40714The Interplay of Economic Shocks and Cultural Practices on Child Marriage: Comparative Evidence from India and Zambia during the COVID-19 PandemicPLOS ONE

Dear Dr. Steinert,

Thank you for submitting your manuscript to PLOS ONE. After careful consideration, we feel that it has merit but does not fully meet PLOS ONE’s publication criteria as it currently stands. Therefore, we invite you to submit a revised version of the manuscript that addresses the points raised during the review process.

As also highlighted in the reviews, you are requested to follow the PLOS One manuscript format.

A rebuttal letter that responds to each point raised by the academic editor and reviewer(s). You should upload this letter as a separate file labeled ’Response to Reviewers’.A marked-up copy of your manuscript that highlights changes made to the original version. You should upload this as a separate file labeled ’Revised Manuscript with Track Changes’.An unmarked version of your revised paper without tracked changes. You should upload this as a separate file labeled ’Manuscript’.

We look forward to receiving your revised manuscript.

Kind regards,

Vandana Dabla

Academic Editor

PLOS ONE

Journal Requirements:

1. Please ensure that your manuscript meets PLOS ONE’s style requirements, including those for file naming. The PLOS ONE style templates can be found at

“German Research Foundation”

“We are grateful for funding received from the German Research Foundation, and World Bicycle Relief.”

“German Research Foundation”

6. We note you have included a table to which you do not refer in the text of your manuscript. Please ensure that you refer to Table 3 in your text; if accepted, production will need this reference to link the reader to the Table.

Reviewers’ comments:

Reviewer’s Responses to Questions

**Comments to the Author**

1. Is the manuscript technically sound, and do the data support the conclusions?

Reviewer #1: Yes

Reviewer #2: Yes

2. Has the statistical analysis been performed appropriately and rigorously? 

Reviewer #1: Yes

Reviewer #2: Yes

3. Have the authors made all data underlying the findings in their manuscript fully available?

Reviewer #1: Yes

Reviewer #2: Yes

4. Is the manuscript presented in an intelligible fashion and written in standard English?

Reviewer #1: Yes

Reviewer #2: No

5. Review Comments to the Author

Reviewer #1: Although the manuscript is well written, there are a few places where either because it seems it was written at the time of COVID-19, or because the description needs better or more explanation, the authors should re-write and improve the descriptions for better understanding by the reader.

The second issue is that the format used for writing is different from the one used by PLOS ONE. It should be the Vancouver style, citing by numbers and without endnotes.

Reviewer #2: This is a comprehensive and important study. Your methods provided a breadth of data. It a useful study to inform policy on future economic shocks and the impact on child marriage.

There are a number of grammatical errors and a lack of citations in some areas:

pg 1 line 43 needs citation

Some areas are vague - eg use of word ’many’ page two line 56

Line 67 page 2 - meaning unclear. Line 68 - vague - what sort of marriage trends? Figures?

Pg 4 line 126 - vague -where is this based. Line 136 - avoid words like ’poorer’ - resource poor?

Pg 5 line 167 - whose awareness?

pg 6 line 170 - evidence of stigma?

Pg 7 line 216 avoid use of words ’developing countries’ - majority world countries?

Line225 - evidence of ’methodological thin’ - also strange expression

Line 248 - why are they conceptual?

249 - make the difference between dowry and brideprice clear. Make clear that Brideprice itself isn’t problematic but it is if it encourages child marriage.

Line 329 avoid the gendered term ’headmasters’

333 - define random walk procedure - where has this been done previously? Cite?

339 - unclear sentence

558 - typo - could?

More generally, I would like to see the conceptual framework referred to at the onset of the paper referred to in greater depth in the discussion.

6. PLOS authors have the option to publish the peer review history of their article (what does this mean?). If published, this will include your full peer review and any attached files.

Reviewer #1: **Yes:** Dr. Alfredo L. Fort

Reviewer #2: **Yes:** Dr Sally Bashford-Squires

---

## [Author Response · Author response to Decision Letter 1]

14 Jan 2026

Our Response Letter is attached as a separate document.

---

## [Decision Letter · Decision Letter 1]

6 May 2026

The Interplay of Economic Shocks and Cultural Practices in Child Marriage: Comparative Evidence from India and Zambia during the COVID-19 Pandemic

PONE-D-25-40714R1

Dear Dr. Steinert,

We’re pleased to inform you that your manuscript has been judged scientifically suitable for publication and will be formally accepted for publication once it meets all outstanding technical requirements.

An invoice will be generated when your article is formally accepted. Please note, if your institution has a publishing partnership with PLOS and your article meets the relevant criteria, all or part of your publication costs will be covered. Please make sure your user information is up-to-date by logging into Editorial Manager at Editorial Manager® and clicking the ‘Update My Information’ link at the top of the page. For questions related to billing, please contact billing support.

Kind regards,

Kasi Eswarappa

Academic Editor

PLOS One

Additional Editor Comments (optional):

Our best wishes.

Reviewers’ comments:

Reviewer’s Responses to Questions

**Comments to the Author**

1. If the authors have adequately addressed your comments raised in a previous round of review and you feel that this manuscript is now acceptable for publication, you may indicate that here to bypass the “Comments to the Author” section, enter your conflict of interest statement in the “Confidential to Editor” section, and submit your "Accept" recommendation.

Reviewer #1: All comments have been addressed

2. Is the manuscript technically sound, and do the data support the conclusions?

Reviewer #1: Yes

3. Has the statistical analysis been performed appropriately and rigorously? 

Reviewer #1: Yes

4. Have the authors made all data underlying the findings in their manuscript fully available?

Reviewer #1: Yes

5. Is the manuscript presented in an intelligible fashion and written in standard English?

Reviewer #1: Yes

6. Review Comments to the Author

Reviewer #1: The authors have carefully and with detail addressed the comments made by reviewers. I believe the article is ready for publication, however looking at a few very minor areas that the publishers will review at such time. Thanks.

7. PLOS authors have the option to publish the peer review history of their article (what does this mean?). If published, this will include your full peer review and any attached files.

Reviewer #1: **Yes:** Dr. Alfredo L Fort, MD, MSc, PhD

---

## [Editor Report · Acceptance letter]

PONE-D-25-40714R1

PLOS One

Dear Dr. Steinert,

I’m pleased to inform you that your manuscript has been deemed suitable for publication in PLOS One. Congratulations! Your manuscript is now being handed over to our production team.

Lastly, if your institution or institutions have a press office, please let them know about your upcoming paper now to help maximize its impact. If they’ll be preparing press materials, please inform our press team within the next 48 hours. Your manuscript will remain under strict press embargo until 2 pm Eastern Time on the date of publication. For more information, please contact onepress@plos.org.

Kind regards,

on behalf of

Dr. Kasi Eswarappa

Academic Editor

PLOS One